# Synergistic Improvement in Coating with UV Aging Resistance and Anti-Corrosion via La-Doped CeO$_2$ Powders

**Ang Tian [1], Tengda Ma [2], Xiaoguo Shi [1,3], Dixiang Wang [1], Wenyuan Wu [4], Chuangwei Liu [1] and Wenli Pei [2,*]**

1  Liaoning Provincial Key Laboratory of Metallurgical Resources Circulation Science, Northeastern University, Shenyang 110819, China; Tiana@mail.neu.edu.cn (A.T.); neusxg@163.com (X.S.); dixiang_wang@163.com (D.W.); chliua@dtu.dk (C.L.)
2  College of Materials Science and Engineering, Northeastern University, Shenyang 110819, China; 18341358086@163.com
3  School of Environmental Science and Engineering, Qilu University of Technology (Shandong Academy of Sciences), Jinan 250353, China
4  Institute of metallurgy, Northeastern University, Shenyang 110819, China; wuwy@smm.neu.edu.cn
*  Correspondence: peiwl@atm.neu.edu.cn; Tel.: +86-024-83691573

**Abstract:** Benefitting from a suitable band gap, ceria is an excellent material for UV shielding. By solid solution doping and specific micromorphology, its band gap can be effectively controlled. In this paper, ceria doped with lanthanum via oxalate precipitation is combined with a high-temperature roasting process. The properties of the prepared samples are characterized by UV–Vis diffuse reflectance spectroscopy (DRS), Raman, XRD, FESEM and XPS. The absorption threshold of materials is clearly red-shifted in the ultraviolet band, which originates from the electron-phonon generation. To further reveal the mechanism, the density function theory calculation (DFT) is implemented to study the influence of lanthanum concentrations on ceria's band gap. It is demonstrated that the band gap can even be narrowed to 2.97 eV by optimizing the sintering temperature and lanthanum-doped concentration. To investigate its improved anti-aging properties under ultraviolet rays, different amounts of 5% lanthanum-doped ceria is mixed with an Al-based coating and then coated on the Q235 steel. Combined with an ultraviolet light irradiation experiment and electrochemical test technology, the corrosion resistance of the modified coatings is evaluated. The coating with 20% La-doped ceria provides the best corrosion resistance performance.

**Keywords:** ceria oxide; lanthanum; density function theory calculation; corrosion resistance

## 1. Introduction

Cerium is one of the lanthanide elements, and the reserves in the earth's crust are rich [1]. Its special 4f electronic track outside the nucleus leads to two types of valences: quadruples and rivals [2]. Correspondingly, the oxides include cerium oxide and trioxide. Both cerium oxide crystal structures are facial cubes, and the oxygen atoms in the facial position can release and absorb oxygen atoms with changes of oxygen partial pressure in the external environment [3,4]. Thus, the chemical molecular formula of cerium oxide is Ce$_7$O$_{12}$ in most cases [5]. Cerium oxide is an ideal ultraviolet shielding material, and since ultraviolet rays can induce the degradation of organic substances, cerium oxide is often added as an additive to coatings to alleviate the coating degradation caused by them [6,7]. Saeia et al. demonstrated that cerium oxide can improve the corrosion resistance of a coating by reducing the concentration of hydrogen and oxygen in a zinc-containing coating [8]. Therefore, cerium oxide has proved its potential in modifying coating properties based on its characteristics of shielding ultraviolet rays and enhancing corrosion resistance.

However, the wide band gap of cerium oxide (3.2 eV) limits its application as an ultraviolet shielding material [9,10]. To date, the doping of metal or non-metal elements, including C, N, S, and Zr, has been widely reported to modify the band gap of cerium

oxide [1,10], including La- or Zr-doped cerium oxide materials, which have been used in solar cell modules [11,12]. Moreover, it has also been proven that oxygen vacancy doping in cerium oxide can induce the red shift of the absorption threshold under ultraviolet and visible bands [13]. The band gap of cerium oxide can be decreased to the range of 2.63–2.75 eV by introducing doping energy levels. In addition, La doping can also reduce the band gap, as it causes the splitting of the crystal field [14]. In turn, the reduced band gap directly induces the red shift of the absorption threshold of cerium oxide in the ultraviolet and visible bands [15].

The ultraviolet band covers 200–400 nm, which can be absorbed by modified cerium oxide. Therefore, with modified cerium oxide introduced into the coating, the effect of anti-ultraviolet degradation can be realized. In this paper, two kinds of cerium oxide materials doped with oxygen vacancies and lanthanum were prepared. It was found that a lower sintering temperature can produce a certain amount of oxygen vacancies in the microstructure of cerium oxide, while the appropriate doping of lanthanum can also introduce doping energy levels. Both methods can reduce the band gap of cerium oxide. The optical properties of pristine and La-doped ceria were then studied through density functional theory (DFT), and the decrease in the band gap was explained based on the density of states. Finally, a lanthanum-doped cerium oxide material was selected and added to an aluminum-based coating, which was then coated on a stainless steel surface. The corrosion resistance of the coating was investigated under strong ultraviolet irradiation.

## 2. Experiment and Calculations

The $CeO_2$ powders were synthesized using the carbonate precipitation method. Firstly, the cerium chloride solution (300 g/L) and the saturated sodium carbonate solution were prepared. Those two solutions (each of 500 mL) were poured into a 1000 mL beaker dropwise with two separating funnels. The whole process was maintained between 40 °C and 45 °C with water bath heating and stirring. The precipitation reaction terminated when the pH of the mixed solution reached 7.5, and the stirring process lasted for 40 min to ensure complete precipitation under 950 °C [16]. The doping cerium samples were fabricated using the oxalate precipitation process. A cerium chloride solution and a lanthanum chloride solution were prepared with concentrations of 300 g/L [17], and three kinds of cerium powders with lanthanum at different concentrations (5 wt.%, 10 wt.% and 15 wt.%) were fabricated using the oxalate precipitation process [18,19]. The detailed process is as following: (1) the corresponding mixed rare earth chloride solutions were prepared according to the lanthanum doping concentration; (2) 200 mL of the mixed solution was added to the beaker with stirring; (3) 400 mL of the saturated sodium oxalate solution was added to the beaker in 30 min intervals. It should be noted that the whole process was kept at 45 °C by the heating device. The sediment was gathered by a vacuum extraction filter and calcined in the muffle furnace (Shenyang Great Wall Industrial Electric Furnace Plant, Shenyang, China) at 950 °C for 2 h.

For the evaluation of the corrosion resistance of the coatings, different amounts of cerium oxide powders doped with lanthanum were added into the commercial aluminum-based coatings (Shenyang Shi Hangda Technology Co., Ltd., Shenyang, China). The substrate was first mounted in a Teflon holder (Aruibok, Dongguan, China), and then the lower surface of the substrate was allowed to contact the sol for 30 s. The substrate was then rotated horizontally at a rotation rate of 800 rpm for 25 s using a spinner (Aruibok, Dongguan, China). The prepared CMP sol was coated onto cp-Ti substrates by the dip-spin coating at 8000 rpm for 1 min. Before the process of the addition, lanthanum-doped cerium powders were ultrasonic dispersed in anhydrous ethanol, and the saline coupling agent was then dispersed in anhydrous ethanol with stirring. The dispersed cerium oxide and silane coupling agent were added into anhydrous ethanol based on the mass ratio of 19:1, and the cerium oxide powders were collected after being centrifuged with 1000 rpm and dried in a vacuum drying oven at 200 °C for 2 h (XMTD-8222, Jing Hong Experimental Equipment Co., Ltd., Shanghai, China). The cerium oxide powders modified by the silane

coupling agent were added into the commercial aluminum-based coatings in the proportion of 1:4 with stirring. The metal substrates used were Q235 steel with the dimensions of 1 cm × 1 cm × 0.2 cm and were treated by sanding before being coated by the aluminum-based coatings, with and without cerium oxide powder. The metal substrates were coated by a dipping process and dried for 2 h.

In this study, a scanning electron microscope (SEM, ZEISS ULTRA, Carl Zeiss, Jena, Germany) was used to characterize the micromorphology of the cerium oxide powders. The crystalline structure was analyzed by X-ray diffraction (XRD, Philips X'Pert PRO, Philips, Amsterdam, The Netherland) with a Cu Kα radiation source, and the chemical composition was measured by X-ray photoelectron spectroscopy (XPS, ESCALAB250, Thermo, Waltham, MA, USA) using Al 2 mm Kα monochromatic radiation as an exciting source. A Raman spectrometer (HR800, Horiba Jobin Yvon, Longjumeau, France) was used to study the structure of the cerium oxide. A UV–Vis spectrometer (Shimadzu, UV2550, Shimadzu, Kyoto, Japan) was employed to study the absorption threshold value of the powders. In this study, a conventional 12-atom cubic unit cell of $CeO_2$ was used as a simplified $CeO_2$ model, and the oxygen vacancy model was created by taking an O atom from the lattice, which is based on the $CeO_2$ experimental structure without further geometry optimization. The computations were performed using the Perdew–Burke–Ernzerhof exchange correlation functional [20]. The 500 eV energy cutoff was adopted, and the Brillouin zone was sampled with 7 × 7 × 7 k-points using the Monkhorst–Pack scheme grid for geometry optimization and self-consistent calculations [21,22].

An ultraviolet irradiation experiment and electrochemical testing technology were employed to evaluate the corrosion resistance of the aluminum-based coatings. The aluminum-based coatings were used as a control sample, and the added coatings of lanthanum-doped cerium was studied using the polarization curve and impedance spectrum. An ultraviolet light with 500 W power was employed, and a mercury lamp was the light source. An electrochemical workstation (Shanghai Chenhua CHI 650C) was employed in this study, and the dynamic potential scanning range was −1.2–1.5 V, the scanning rate was 5 mV/s, the frequency was 2 Hz, and the electrolyte was sodium chloride solution with a concentration of 3.5 wt.%.

## 3. Results

The XRD results of pristine and La-doped $CeO_2$ nanopowders are shown in Figure 1. The diffraction plane of ceria at (111), (200), (220), (311), (222), (400), (331) and (420) could be observed in the spectrum of ceria with four different lanthanum concentrations, namely, 0 wt.%, 5 wt.%, 10 wt.% and 15 wt.%. It can be clearly observed that the fine powders are successfully synthesized in our work as a result of the wide diffraction peaks. It can be seen that with the increase in lanthanum concentration, the position of diffraction peak shifts to the small angle [23,24]. The radiuses of lanthanum and cerium ions are 1.061 and 0.94 Å, respectively. Lanthanum doping of the lattice of $CeO_2$ could be defined as the unequal exchange, which results in oxygen vacancy and impairment of lattice integrity. The crystal structure of ceria is a face-centered cubic octahedron, and the ceria lattice is a composite lattice structure. The gap size of face-centered cubes is constructed by cerium and oxygen atoms. Both sizes are smaller than the diameter of the lanthanum ion; thus, the crystal structure of lanthanum-doped ceria is a replacement solid solution. The replacement of lanthanum in the solid solution might lead to the increase in the lattice constant, crystal plane spacing and lattice expansion. As the XRD spectrum showed, the diffraction plane shifted to the smaller angle, which agrees with the brag equation $2d\sin\theta = n\lambda$. The crystal structure parameters are shown in Table 1, and the relevant calculation results further prove that the crystal structure of cerium oxide with lanthanum doping is a replacement solid solution.

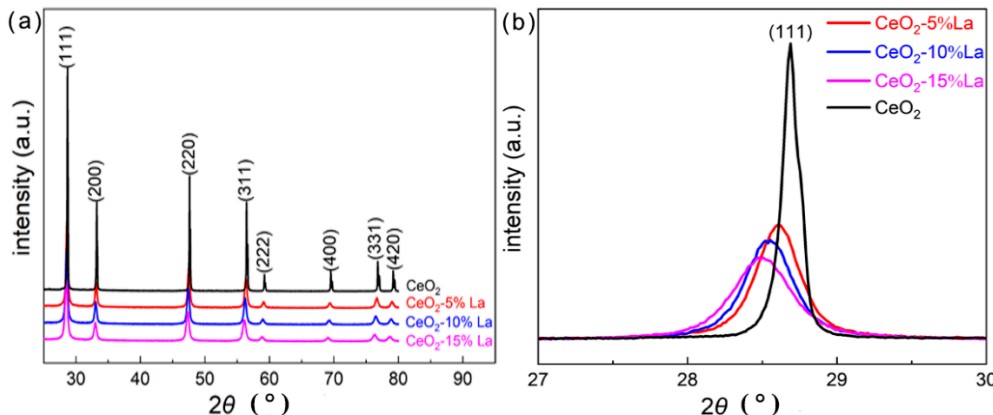

**Figure 1.** (**a**) XRD patterns of cerium oxide doped with lanthanum at different concentrations. (**b**) Magnified peaks at 28.6° of 2θ.

**Table 1.** XRD parameters of cerium oxide doped with lanthanum at different concentrations.

| La-Doped $CeO_2$ (%) | Degree | d/Å | (hkl) | FWHM | Grain Size/nm | Lattice Constant/Å |
|---|---|---|---|---|---|---|
| 5 | 28.53 | 3.13 | 111 | 0.16 | 49.43 | 5.42 |
| 10 | 28.67 | 3.13 | 111 | 0.16 | 49.47 | 5.42 |
| 15 | 28.63 | 3.13 | 111 | 0.16 | 50.04 | 5.43 |

The morphologies of $CeO_2$ powder with doped different La concentrations are shown in Figure 2. The uniform and flat pure $CeO_2$ powder was prepared using the carbonate precipitation method with an average size of 120 nm. The particle size decreased when the La atom was introduced into the pure materials, and the average size of nanoparticles was about 50 nm. There were no obvious changes to average size with different concentrations of La. Therefore, the results indicate that introducing the La dopant decreases the $CeO_2$ lattice size. For further confirmation of the crystal structure of the lanthanum-doped ceria, XPS was employed, and the results are shown in Figures 3 and 4. The XPS spectrum of cerium mainly includes eight characteristic peaks, of which two belong to the trivalent ceria ion and the remainder belong to the cerium ion. The peaks of the ceria ion are located at 885.1 and 903.3 eV [25]. The transformation between ceria and cerium frequently occurs to neutralize the excess negative charges caused by the substitution of Ce(III) for Ce(IV), but the oxygen ions can migrate to the outside of the crystal and result in the formation of oxygen vacancies, which maintains the electric neutrality of the crystal. The concentration of the oxygen vacancies depends on the ceria ions, and the related concentrations were calculated [26]. Based on the calculation results, the lowest concentrations of oxygen vacancies and ceria ions in the $CeO_2$ doped with 5% lanthanum were 2.93% and 11.73%, respectively. The results above, based on XRD, Raman and XPS, indicate that lanthanum was doped in the crystal lattice of cerium oxide in the form of a solid solution. There is no phase formation of lanthanum oxide in the lanthanum-doped cerium oxide sample, so the energy level structure of cerium oxide forms a doped energy level instead of a heterojunction structure composed of lanthanum oxide and cerium oxide.

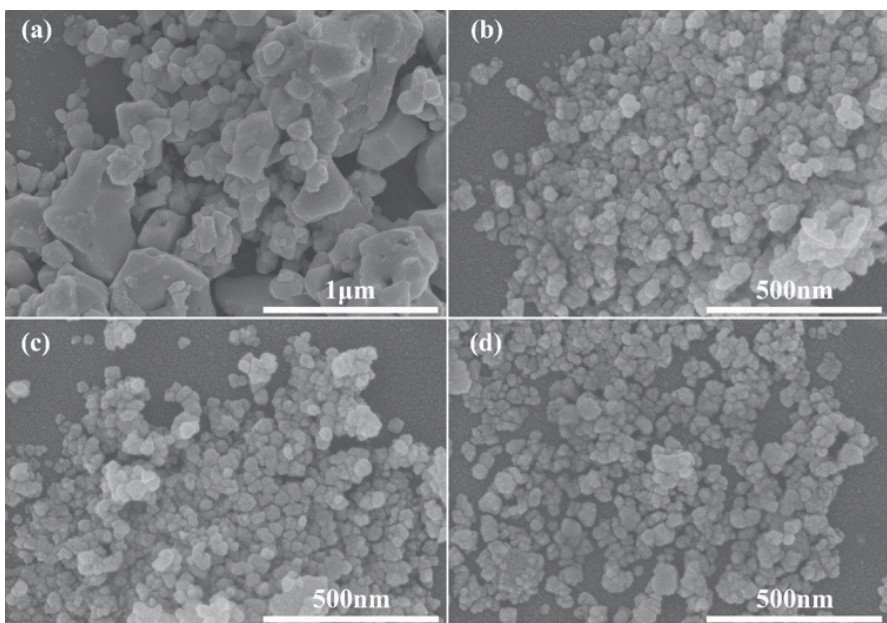

**Figure 2.** FESEM micrograph of cerium oxide doped with lanthanum at different concentrations: (**a**) 0 wt.%, (**b**) 5 wt.%, (**c**) 10 wt.% and (**d**) 15 wt.%.

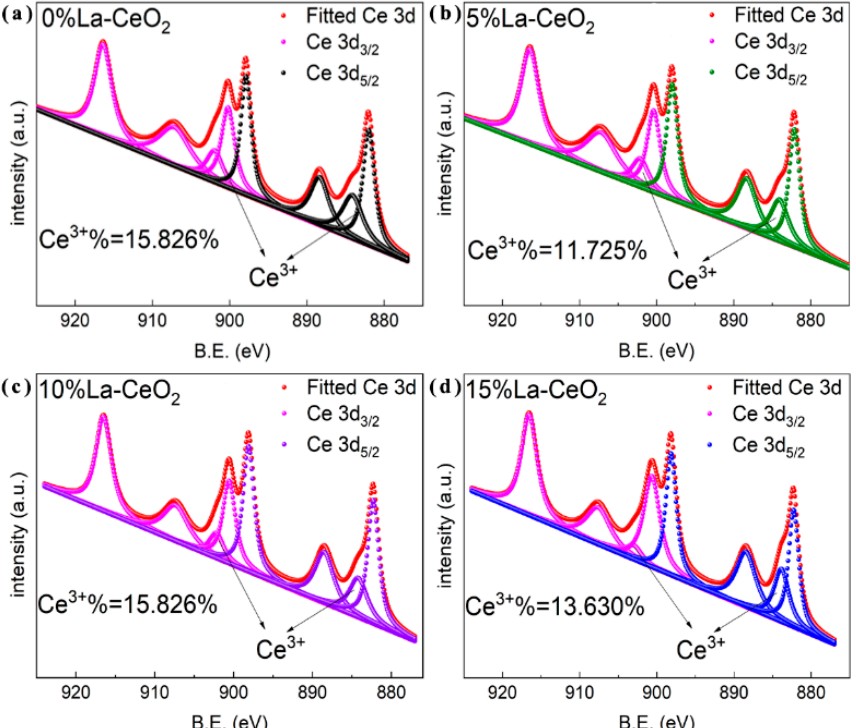

**Figure 3.** XPS of cerium in cerium oxide doped with lanthanum at different concentrations: (**a**) 0% La-CeO$_2$, (**b**) 5% La-CeO$_2$, (**c**) 10% La-CeO$_2$ and (**d**) 15% La-CeO$_2$.

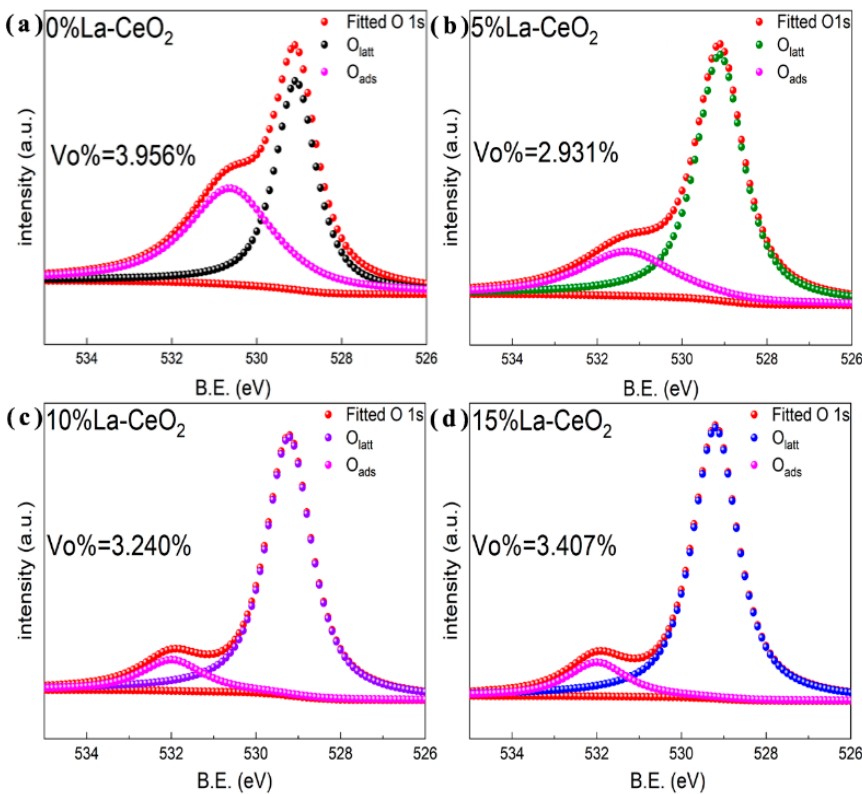

**Figure 4.** XPS of oxygen in cerium oxide doped with lanthanum at different concentrations: (**a**) 0% La-CeO$_2$, (**b**) 5% La-CeO$_2$, (**c**) 10% La-CeO$_2$ and (**d**) 15% La-CeO$_2$.

The Raman spectra of samples are presented in Figure 5, and the pure CeO$_2$ holds the highest intense peak at about 470 cm$^{-1}$ which corresponds to the symmetrical stretching of the Ce–O bond. A slightly Raman-active peak is located at 395 cm$^{-1}$, which is ascribed to La$_2$O$_3$ [23,27]. As we know, the pattern of CeO$_2$ is very sensitive to any disorder in the presence of La atoms, and the main peak shifts to a lower angle (460 cm$^{-1}$) with weaker strength, becoming more asymmetric when the CeO$_2$ powder is doped with La. It is likely that Ce$^{3+}$ ions and La in the samples are responsible for the changes in the Raman scattering as supported by the XPS data.

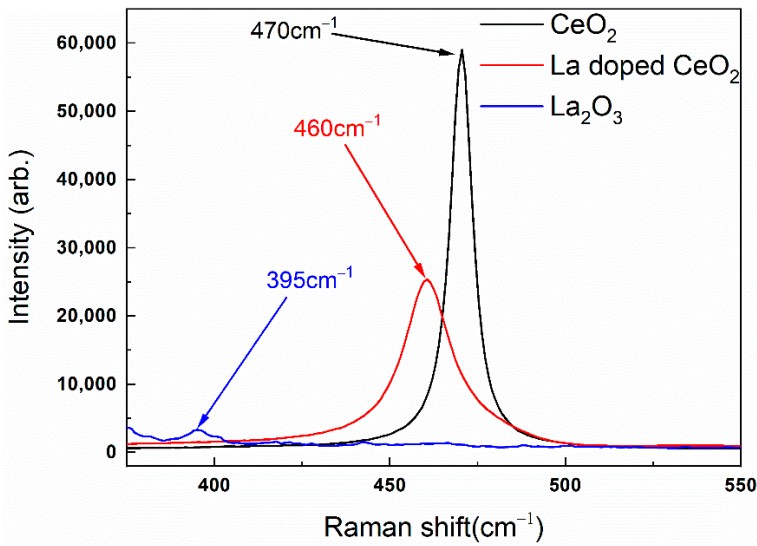

**Figure 5.** Raman spectrum of CeO$_2$ (black line), La-doped CeO$_2$ (red) and La$_2$O$_3$ (blue).

The UV–Vis adsorption spectra of La-doped $CeO_2$ samples were studied by theory and experimental calculation, and the results are displayed in Figure 6. The calculated optical adsorption spectra of cerium oxide doped with lanthanum are shown in Figure 6a. It is noted that the absorption edges of La-doped $CeO_2$ show an obvious red shift, especially at the edges where 5% La-$CeO_2$ reaches 450 nm (visible light area). From around 500 nm to the near-infrared (NiR) range, the absorption of all samples does not occur, so the spectra are flat. The ultraviolet absorption capability of pure cerium oxide benefits from the charge transfer from $O^{2-}$ to $Ce^{4+}$ ions. In this study, the absorption coefficient was calculated using the formula $\alpha = (-1/d) \times \ln(T)$, where d is the sample thickness and T is the transmittance. The calculation result (Figure 6a) and experimental result (Figure 6b) show a similar trend, especially the adsorption peak of 5% La-$CeO_2$ located in the VIS area. As Figure 6b shows, the band gaps of cerium oxide with different lanthanum doping are 3.2, 2.97, 3.01, and 3.04 eV. The band gap first decreases with an increase in the concentration of La doping in the cerium oxide and reaches the minimum value (2.97 eV) at 5% La doping; then, it shows an increasing trend. Therefore, the 5% La doping sample provides the best UV-shielding performance.

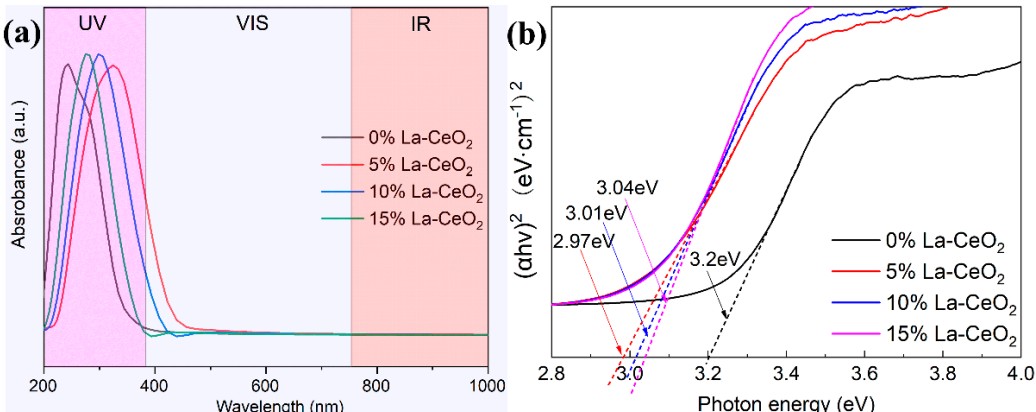

**Figure 6.** (**a**) Calculated optical adsorption spectra of cerium oxide doped with different lanthanum concentrations (0%, 5%, 10%, and 15%); (**b**) forbidden band width of above four samples.

To investigate the UV aging resistance, the 5% La-$CeO_2$ mixed coatings were coated onto the stainless steel substrate, and the concentrations of the powder in the coating were 10 wt.%, 20 wt.% and 30 wt.%. It can be seen from Figure 7g–h that the average thicknesses of the control coating ($CeO_2$ coating) and 20% La-$CeO_2$ coating were about 57.67 and 56.46 μm, respectively. These similar thicknesses indicate the good thickness repeatability of the coating preparation process. Figure 7 illustrates the FESEM results of the stainless steel substrate coated with a $CeO_2$ layer with La doping. In Figure 7a, the whole surface of the sample was kept perfectly uniform, while some white and bright flocs appeared after the La was introduced into the pure $CeO_2$ powder. To further confirm the component of the flocs, EDS analysis was employed to the circle area, and the results are summarized in Figure 7c–f. It was observed that La was successfully and uniformly doped into the coating. Figure 7g,h present the cross-sectional images of control samples with a pure $CeO_2$ coating and a 20% La-$CeO_2$ coating after testing. The coatings still showed good adhesion with the substrates after testing. However, the microstructure of coating layers was considerably different. The control sample was relatively porous compared to the dense 20% La-$CeO_2$ coating. Thus, the corrosion resistance of the layer was enhanced by introducing La into the $CeO_2$ coating.

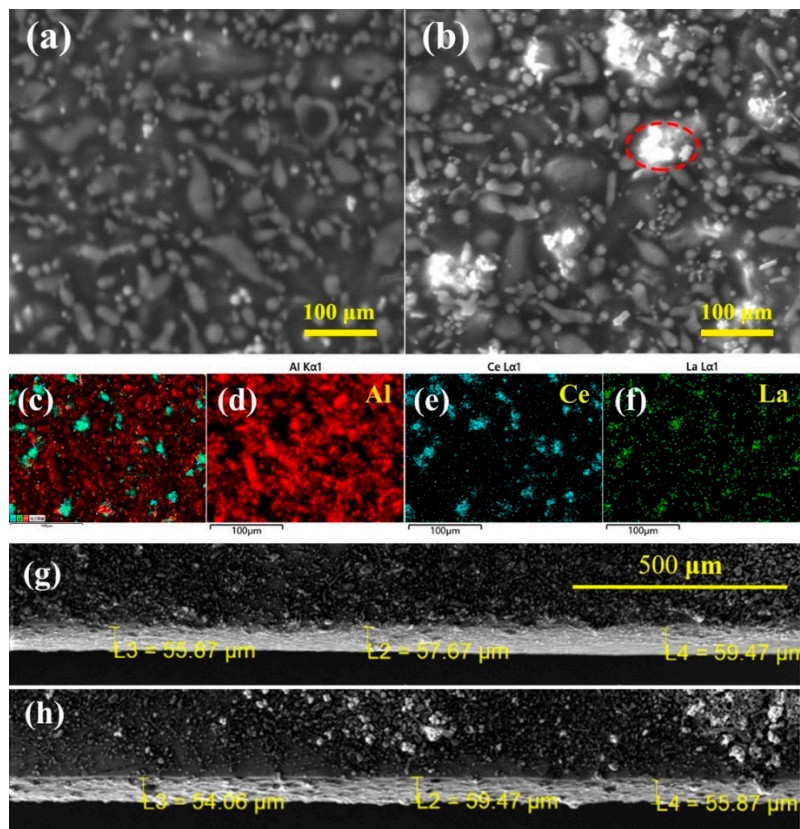

**Figure 7.** FESEM micrographs of (**a**) control coating, (**b**) 20% La-CeO$_2$ coating, (**c–f**) EDS mapping of 20% La-CeO$_2$ coating, cross-sectional view of (**g**) control coating and (**h**) 20% La-CeO$_2$ coating.

To explore the effect of ultraviolet light irradiation on coating performance, the CeO$_2$ coating and 20% La-CeO$_2$ coating were placed under a high-pressure mercury lamp for 12 h with an ultraviolet wavelength of 350 nm and irradiation distance of 20 cm. The polarization curve (Figure 8a) and impedance spectrum (Figure 8b–d) were employed to study the corrosion resistance of coatings after 12 h UV irradiation. In the equivalent circuit, RS represents solution resistance, CPE is equivalent to capacitance, and RC represents the interface resistance between the solution and the coating. As Figure 8 and Table 2 show, compared with the CeO$_2$ coating without La doping, the lower self-corrosion current density (0.0232 μA·cm$^2$) of the 20% La-CeO$_2$ coating implies a better anti-UV irradiation performance. In addition, the 20% La-CeO$_2$ coating has a bigger radius in the Nyquist plot and a higher impedance value (6.2 Ω·cm$^2$) in the low-frequency region of the Bode plot than the CeO$_2$ coating, which indicates ideal anti-UV resistance.

**Table 2.** Electrochemical data of La-doped CeO$_2$ coating after UV irradiation for 12 h.

| Coating | Ecorr/V | Icorr/μA·cm$^{-2}$ | Low-Frequency Region Log (\|Z\|/Ω·cm$^2$) |
|---|---|---|---|
| CeO$_2$ | −0.52 | 0.51 | 4.75 |
| 10% La-CeO$_2$ | −0.56 | 0.21 | 5.18 |
| 20% La-CeO$_2$ | −0.54 | 0.02 | 6.20 |
| 30% La-CeO$_2$ | −0.54 | 0.58 | 4.68 |

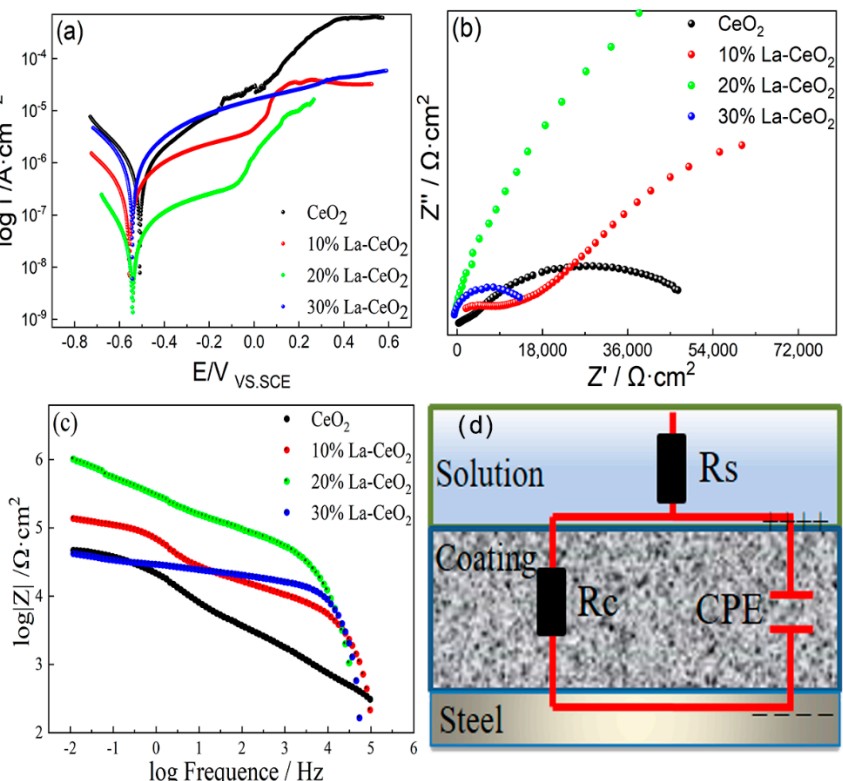

**Figure 8.** Electrochemical analysis of La-doped $CeO_2$ coatings after UV irradiation for 12 h. (**a**) Polarization curves, (**b**) Nyquist plots, (**c**) Bode plots and (**d**) equivalent circuit.

Based on the UV irradiation and electrochemical experiment, the effect of coatings modified by cerium oxide on corrosion resistance could be attributed to La doping. The aging and degradation of coatings are caused by the synergy of light and heat. Ultraviolet sunlight can induce the breaking of double bonds in organic coatings, resulting in the generation of hydroxyl radicals and hydroperoxide. These groups accelerate the breaking of molecular bonds and lead to photodegradation [28]. The aging of the coating makes the stress concentration area of the coating more prone to generating pores and corrosion microcracks [29]. Water molecules, oxygen molecules and chloride ions invade the interface between the coating and substrate, which can cause corrosion. Cerium oxide doped with lanthanum in the coating absorbs the ultraviolet light and converts it into a short wavelength electromagnetic wave; thus, the degradation of coatings can be avoided.

## 4. Discussions

In this study, we proposed the use of La-doped $CeO_2$ powder for UV shielding via the carbonate precipitation method. The wide diffraction peaks indicate that the grains of the samples are very fine due to the introduced La atoms. The absence of the peak near 395 cm$^{-1}$ corresponding to $La_2O_3$ indicates the incorporation of $La^{3+}$ into the $CeO_2$ lattice. The UV–Vis adsorption spectra of La-doped $CeO_2$ samples were studied by theory and experimental calculation. As the concentration of the lanthanum dopant increased from 0 to 15%, the adsorption band red-shifted progressively from 275 to 345 nm. The red shifts indicate that optical band gaps are correlated with the dopant concentrations, and the band gap value of the materials decreased from 3.20 to 2.97 eV, especially the absorption edge of the sample with an La concentration of 5% that reached 450 nm. During the investigation of the UV aging resistance of La-doped $CeO_2$ powder, it was observed that the Al-based coating presented the best performance when the concentration of 5% La-$CeO_2$ was as high as 20%. It is known that the corrosion resistance of the anode is closely related to the morphology and thickness of the oxide layer, where a denser and thicker surface oxide layer leads to better corrosion resistance. EIS analysis was used to study the anticorrosive

performance of the epoxy coating on Q235 carbon steel substrates treated with a dopant at different concentrations. For the 20% La/$CeO_2$ powders coated onto a Q235 carbon steel sample, the corrosion current density of the sample treated with La-doped $CeO_2$ powder increased by one order of magnitude compared to the current density of the untreated material. This performance is primarily related to the UV aging resistance of the introduced La. The high-frequency (HF) semicircle was attributed to coating pore impedance Rp, while the low-frequency semicircle was the impedance response associated with the corrosion reaction occurring at the interface through defects and pores in the coating. The Nyquist plots show that the diameters of these capacitive arcs increased with the increase of La doping. This behavior indicates that impedance of the steel sample against corrosion increased according to the amount of La in the coating powder. The corresponding Bode plots confirm that the addition of La causes red shift against most ultraviolet rays. In addition, it shows that the bode phase angle was enhanced with the concentration of La investigated. Therefore, the 20% La/$CeO_2$ powders mixed into the coatings had a bigger radius in the Nyquist plot and a higher impedance value (6.2 $\Omega \cdot cm^2$) in the low-frequency region of the Bode plot than the $CeO_2$ coating, which indicates ideal anti-UV resistance.

## 5. Conclusions

In summary, four different cerium oxide powders with different concentrations of La as a dopant were prepared using the oxalate precipitation process under 950 °C, and all samples were characterized by XRD, XPS and FESEM in this work. The UV–visible diffuse reflectance spectra indicate that the La-doped $CeO_2$ has optical capability in nearly the whole range of the visible light spectrum. From the experimental and DFT studies, the results show that the band gap of samples decreased by introducing an La atom into the lattice of a $CeO_2$ 5% sample, thereby providing the minimum value (2.97 eV). The UV–Vis adsorption spectra presented a red-shift phenomenon in the presence of the La dopant. Finally, we used EIS and salt spray testing to evaluate the anti-aging property of La-doped $CeO_2$, which was mixed with an Al-based coating and despotized on Q235 steel substrate in 3.5 wt.% NaCl solution. The corrosion resistance of the substrate was significantly enhanced by the introduction of the La dopant in the $CeO_2$ powder.

**Author Contributions:** Conceptualization, A.T.; project administration, T.M. and X.S.; methodology, D.W.; software, C.L.; data curation, W.W.; funding acquisition, W.P. All authors have read and agreed to the published version of the manuscript.

**Funding:** National Natural Science Foundation of China (Grant Nos. 51871045 and 52071070), and Basic Scientific Research Foundation of Central College (Grant Nos. N2125032).

**Institutional Review Board Statement:** Not applicable.

**Informed Consent Statement:** Not applicable.

**Conflicts of Interest:** The authors declare no competing financial interests.

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
