# Peer review of "Synergistic Improvement in Coating with UV Aging Resistance and Anti-Corrosion via La-Doped CeO2 Powders"

_coatings, doi:10.3390/coatings11091095_

Round 1
Reviewer 1 Report
Cerium oxide improves the resistance of the coatings to UV radiations, but its broad bandgap limits the application field. Lanthanum (La) can reduce the bandgap.
In this paper, two kinds of cerium oxide powder doped with oxygen vacancies and lanthanum were used as blends in various ratios for the aluminum coating of Q235 steel. The optical properties of pristine and coated Q235 samples were studied, besides their corrosion resistance.
The paper is original and the research is well conducted. Unfortunately, the presentation is not the best, with important information being omitted. The repeatability of the tests is compromised at the moment. To bring the paper to the requested publishing standards, the authors ought to consider the next suggestions:
Major revision points:
- The discussion section is missing. The results presented in each figure must be discussed. Compare your results against those found in the literature and emphasize the originality and the contribution of your paper.
- Specify the main properties of the coating powder, Q235 steel, the deposition parameters of the coating process, and the thickness of the deposited coatings.
- As it can be seen, the solution for best UV resistance is not the same as for best corrosion resistance. Discuss this aspect in the Discussion section, propose the best compromise and compare the performances against the existing results from the literature.
- In the Conclusions section, emphasize the optimum solution for both the best UV and corrosion resistance.
Minor revision points:
- Place the Figures and Tables immediately after mentioning them in the paper, and not in a separate section.
- For each employed equipment, the manufacturer, the country, and the city have to be mentioned.
- Cite the references before the full stop of the involved phrase.
Author Response
- The discussion section is missing. The results presented in each figure must be discussed. Compare your results against those found in the literature and emphasize the originality and the contribution of your paper.
A: Thanks for your suggestion, we added the discussion part in our new manuscript.
Discussions
We propose La-doped CeO2 powder for UV shielding by the carbonate precipitation method. The wide diffraction peaks indicate that the grains of the samples are very fine by introduced La atoms. The absence of the peak near 395 cm-1 correspondings to La2O3 indicates the La3+ incorporation into the CeO2 lattice. The UV-VIS absorption spectra of La-doped CeO2 samples are studied by theory and experimental calculation. The results show that absorption edges of La-doped CeO2 show obvious redshift, especially the edges of 5% La contents sample reaches to 450 nm. And EIS analysis is used to study the anticorrosive performance of the epoxy coating on Q235 carbon steel substrates treated with different doped weights. The 20% La/CeO2 powders are mixed into coatings, which has a bigger radius in the Nyquist plot and higher impedance value (6.2 Ω.cm2) in the low-frequency region of the Bode plot than the CeO2 coating, which indicated the ideal anti UV resistance.
- Specify the main properties of the coating powder, Q235 steel, the deposition parameters of the coating process, and the thickness of the deposited coatings.
A: Thanks for your suggestion. We added more details of the coating process and thickness of coatings in our new manuscript.
The substrate was first mounted in a Teflon holder and then the lower surface of the substrate was allowed to contact the sol for 30 s. The substrate was then rotated horizontally at a rotation rate of 800 rpm for 25 s using a spinner. The prepared CMP sol was coated onto cp-Ti substrates by the dip-spin coating at 8000 rpm for 1min.
It can be seen from Figure 7 (g) and (h), the average thicknesses of control coating (CeO2 coating) and 20%La-CeO2 coating are about 57.67 μm and 56.46 μm, respectively. A similar thickness indicates the good thickness repeatability of the coating preparation process.
- As it can be seen, the solution for best UV resistance is not the same as for best corrosion resistance. Discuss this aspect in the Discussion section, propose the best compromise and compare the performances against the existing results from the literature.
A: Thanks for your suggestion. Please check our discussion.
- In the Conclusions section, emphasize the optimum solution for both the best UV and corrosion resistance.
A: Thanks for your suggestion, and the following part was added in our paper.
We use EIS and salt spray testing to evaluate the anti-aging property of La-doped CeO2 coatings, which is mixed with Al based coating and despotized on Q235 steel substrate in 3.5 wt.% NaCl solution.
Minor revision points:
- Place the Figures and Tables immediately after mentioning them in the paper, and not in a separate section.
A: Thanks for your suggestion. we wrote the paper by the template of Coatings. We will arrange those Figures and Tables by your suggestions after received.
- For each employed equipment, the manufacturer, the country, and the city have to be mentioned.
A: Thanks for your suggestion. Please check our experimental section.
In this study, the scanning electron microscope (SEM, ZEISS ULTRA, Carl Zeiss, Germany) was used for characterizing the micromorphology of the cerium oxide powders. The crystalline structure was analyzed by X-ray diffraction (XRD, Philips X'Pert PRO, Philips, Netherland) with Cu Kα radiation source, and the chemical composition was measured by X-ray photoelectron spectroscopy (XPS, ESCALAB250, Thermo, USA) using Al 2mm Kα monochromatic radiation as an exciting source. The Raman spectrometer (HR800, Horiba Jobin Yvon, France) was used to study the structure of the cerium oxide. The UV-Vis spectrometer (Shimadzu, UV2550, Shimadzu, Kyoto, Japan) was employed for studying the absorption threshold value of the powders.
- Cite the references before the full stop of the involved phrase.
A: Thanks for your work. We changed the position of references, please check the revised manuscript.

Reviewer 2 Report
Please enhanced the quality of figure 7
Please update your references
Author Response
Please enhance the quality of figure 7
A: Thanks for your suggestion. We upload a new Figure 7 in our new revised draft. Because the coating is the organic layer (Figure 7a) which has poor conductivity even after Au is deposited.
Please update your references
A: Thanks for your work. We updated our references, please check the revised manuscript.

Reviewer 3 Report
This article provides both scientific and practical interests in terms of metal surface protection from UV spectra. In general, this approach would be of great interest on many audiences. There are some small suggestions here possibly improve the research. I did not find actual coating layer thickness information in the article. Discussion about micro morphology of a coated layer would be required influence of layer thickness on the structure, the dielectric properties, and so on. Also, it would be better to reconsider some expressions such as ""La dopant decreases the size of CeO^2 powder". (line 146) "powder" is a bit confusion, and would be described as "lattice size". Regarding experimental approach, some coating experiments were applied "dipping". It would be necessary to describe detail condition of the dipping process and/or number of samples used for the experiments. Since dipping process is highly dependent on actual condition, and at least some minimum statistically meaningful sample numbers would add credibility of the result.
Author Response
This article provides both scientific and practical interests in terms of metal surface protection from UV spectra. In general, this approach would be of great interest on many audiences. There are some small suggestions here possibly improve the research.
- I did not find actual coating layer thickness information in the article. Discussion about the micromorphology of a coated layer would be required influence of layer thickness on the structure, the dielectric properties, and so on.
A: Thanks for your review work. The layer thickness is about 57 μm, and we added it and more information about the in our revised manuscript.
It can be seen from Figure 7 (g) and (h), the average thicknesses of control coating (CeO2 coating) and 20%La-CeO2 coating are about 57.67 μm and 56.46 μm, respectively. A similar thickness indicates the good thickness repeatability of the coating preparation process.
- Also, it would be better to reconsider some expressions such as ""La dopant decreases the size of CeO^2 powder". (line 146) "powder" is a bit confusion, and would be described as "lattice size".
A: Thanks for your suggestions. We change the powder into lattice size.
- Regarding experimental approach, some coating experiments were applied "dipping". It would be necessary to describe detail condition of the dipping process and/or number of samples used for the experiments. Since dipping process is highly dependent on actual condition, and at least some minimum statistically meaningful sample numbers would add credibility of the result.
A: Thanks for your suggestions. The substrate was first mounted in a Teflon holder and then the lower surface of the substrate was allowed to contact the sol for 30 s. The substrate was then rotated horizontally at a rotation rate of 800 rpm for 25 s using a spinner. The prepared CMP sol was coated onto cp-Ti substrates by the dip-spin coating at 8000 rpm for 1min. Three times testing was employed for every sample, and the result of manuscript is average value.

Round 2
Reviewer 1 Report
The authors responded to all my questions and amended the paper as requested.
As a recommendation, the section "Results" must be joined with the section "Discussions" in "Results and Discussions", as the current "Discussions" section is too short.
Author Response
A: Appreciate your efforts for our manuscript, we added more words in our discussion section, please check the following part.
- Discussions
We propose La-doped CeO2 powder for UV shielding by the carbonate precipitation method. The wide diffraction peaks indicate that the grains of the samples are very fine by introduced La atoms. The absence of the peak near 395 cm-1 correspondings to La2O3 indicates the La3+ incorporation into the CeO2 lattice. The UV-VIS absorption spectra of La-doped CeO2 samples are studied by theory and experimental calculation. As the lanthanum-doped content increases from 0 to 15%, the adsorption band red-shifts progressively from 275 to 345 nm. The red shifts indicate that optical band gaps are correlated to the doping contents. And the band gap value of the materials decreases from 3.20 to 2.97 eV. Especially, the absorption edge of the 5% La contents sample reaches 450 nm. To investigate the UV aging resistance of La-doped CeO2 powder, and the Al-based coating presents the best performance when the content of 5% La-CeO2 is up to 20% in this coating. It is known that the corrosion resistance of the anode is closely related to the morphology and thickness of the oxide layer; a denser and thicker surface oxide layer leads to better corrosion resistance. And EIS analysis is used to study the anticorrosive performance of the epoxy coating on Q235 carbon steel substrates treated with different doped contents. For the 20% La/CeO2 powders coated Q235 carbon steel sample, the corrosion current density of the sample treated by La-doped CeO2 powder increased by one order of magnitude compared to the current density of the untreated material. This performance is primarily related to the UV aging resistance of La introduced. The high frequency semicircle is attributed to coating pore impedance Rp, while the low frequency semicircle is the impedance response associated with the corrosion reaction occurring at the interface through defects and pores in the coating. The Nyquist plots display the diameters of these capacitive arcs are increasing with La-doped concentration. This behavior indicates that the impedance of the steel sample against corrosion is increased in accordance with the amount of La-doped in the coating powder. The corresponding Bode plots confirm the addition of La causes red shift, which is against most ultraviolet rays. In addition, it observes that the bode phase angle is enhanced with the concentration of La investigated. Therefore, the 20% La/CeO2 powders are mixed into coatings, which has a bigger radius in the Nyquist plot and higher impedance value (6.2 Ω.cm2) in the low-frequency region of the Bode plot than the CeO2 coating, which indicated the ideal anti UV resistance.
